# Endogenous Regulation and Pharmacological Modulation of Sepsis-Induced HMGB1 Release and Action: An Updated Review

**DOI:** 10.3390/cells10092220

**Published:** 2021-08-27

**Authors:** Cassie Shu Zhu, Wei Wang, Xiaoling Qiang, Weiqiang Chen, Xiqian Lan, Jianhua Li, Haichao Wang

**Affiliations:** 1The Feinstein Institutes for Medical Research, Northwell Health, 350 Community Drive, Manhasset, NY 11030, USA; czhu3@northwell.edu (C.S.Z.); xqiang@northwell.edu (X.Q.); wchen6@northwell.edu (W.C.); xlan@northwell.edu (X.L.); jli@northwell.edu (J.L.); 2Donald and Barbara Zucker School of Medicine at Hofstra/Northwell, 500 Hofstra Blvd, Hempstead, NY 11549, USA; 3School of Plant and Environmental Sciences, Virginia Tech, Blacksburg, VA 24061, USA; wwei6@vt.edu

**Keywords:** sepsis, pyroptosis, innate immune cells, antibodies, herbal medicine, acute-phase proteins, hemichannel, inflammasome

## Abstract

Sepsis remains a common cause of death in intensive care units, accounting for approximately 20% of total deaths worldwide. Its pathogenesis is partly attributable to dysregulated inflammatory responses to bacterial endotoxins (such as lipopolysaccharide, LPS), which stimulate innate immune cells to sequentially release early cytokines (such as tumor necrosis factor (TNF) and interferons (IFNs)) and late mediators (such as high-mobility group box 1, HMGB1). Despite difficulties in translating mechanistic insights into effective therapies, an improved understanding of the complex mechanisms underlying the pathogenesis of sepsis is still urgently needed. Here, we review recent progress in elucidating the intricate mechanisms underlying the regulation of HMGB1 release and action, and propose a few potential therapeutic candidates for future clinical investigations.

## 1. Introduction

Microbial infections and resultant sepsis syndromes are the most common causes of death in intensive care units, accounting for approximately 20% of total deaths worldwide [1]. The pathogenesis of sepsis remains poorly understood, but is partly attributable to immune over-activation or immunosuppression propagated by dysregulated innate immune responses to lethal infections [2,3]. Innate immune cells (such as macrophages, monocytes and neutrophils) constitute a front line of defense against microbial infections by eliminating invading pathogens via phagocytosis, and initiating inflammatory responses via various mediators. Upon detection of microbial products such as bacterial endotoxins (lipopolysaccharide, LPS), circulating neutrophils and monocytes immediately infiltrate into infected tissues [4]. After engulfing and killing pathogens, neutrophils exhaust intracellular enzymes and undergo apoptotic cell death. The cell debris of these apoptotic neutrophils are then removed by tissue macrophages (e.g., Kupffer cells, dendritic cells, and glia cells) [5] terminally differentiated from infiltrated monocytes.

Innate immune cells also carry various pattern recognition receptors (PRRs) to recognize distinct classes of molecules shared by a group of related microbes, which are collectively termed “pathogen-associated molecular pattern molecules” (PAMPs). For instance, Toll-like Receptor 2 (TLR2) [6], TLR4 [7] and TLR9 [8], respectively, serve as PRRs for distinct PAMPs such as peptidoglycans, bacterial endotoxins, and microbial un-methylated CpG-DNAs. The engagement of various PRRs by different PAMPs similarly activates innate immune cells to sequentially release early cytokines (such as tumor necrosis factor (TNF) and interferons (IFNs)) and late-acting pro-inflammatory mediators [such as high-mobility group box 1 (HMGB1) and sequestosome 1 (SQSTM1)] [9,10].

HMGB1 is constitutively expressed by most types of cells to maintain a large “pool” of preformed protein in the nucleus, possibly due to the presence of two lysine-rich nuclear localization sequences (NLS) [11]. It carries two internal repeats of positively charged domains (“HMG boxes” known as “A box” and “B box”) in the N-terminus, and a continuous stretch of negatively charged (aspartic and glutamic acid) residues in the C-terminus. These HMG boxes enable HMGB1 to bind chromosomal DNA, and fulfill its nuclear functions to maintain nucleosomal structure and stability, and regulate gene expression [12,13]. Once released, extracellular HMGB1 can bind many endogenous proteins, thereby modulating divergent innate immune responses to lethal infections [13]. To complement two relevant reviews in this Special Issue [14,15], here we summarize recent progress in elucidating the intricate mechanisms underlying both endogenous regulation and pharmacological modulation of LPS-induced HMGB1 release and action, and propose a few potential therapeutic candidates for future clinical investigations.

## 2. Role of Cell-Surface PRRs in the Regulation of Early Pro-Inflammatory Cytokines

Innate immune cells employ cell-surface PRRs such as TLR4 [7] to recognize extracellular bacterial endotoxins (e.g., LPS) in conjunction with a LPS-binding protein (LBP) [16] and a cell-surface co-receptor CD14 [17,18]. Upon capturing LPS, LBP interacts with CD14 to deliver it to a cell-surface receptor, TLR4 [7], thereby triggering immediate production of early cytokines (e.g., TNF and IFN-γ) and subsequent release of late-acting mediators (e.g., HMGB1 and SQSTM1) [9,10]. The critical role of CD14 in the regulation of LPS-induced inflammation was evidenced by: (1) an enhanced sensitivity to lethal endotoxemia in CD14-over-expressing mice [19]; (2) a reduced susceptibility to lethal endotoxemia in CD14-deficient mice [20]; and (3) an abolishment of LPS-induced production of early cytokines (e.g., TNF) in CD14-deficient innate immune cells [21,22]. However, we found that the depletion of CD14 expression only partly attenuated the LPS-induced HMGB1 release [22], suggesting a potential involvement of other CD14/TLR4-independet signaling pathways in the regulation of HMGB1 release.

In a murine model of endotoxemia induced by intraperitoneal administration of LPS, HMGB1 was first detected in the circulation eight hours after endotoxemia, and subsequently increased to plateau levels from 16 to 32 hours [9]. This late appearance of circulating HMGB1 paralleled with the onset of animal lethality from endotoxemia, and distinguished itself from TNF and other early proinflammatory cytokines [23]. Lacking a leader peptide sequence, HMGB1 could not be actively secreted through classical endoplasmic reticulum-Golgi exocytotic pathways [9]. Instead, upon post-translational acetylation or phosphorylation [24,25] of the nuclear localization sequence (NLS) [26,27], nuclear HMGB1 is translocated to the cytosol and sequestered into cytoplasmic vesicles [11,24,28,29]. These cytoplasmic HMGB1 vesicles could then be secreted into the extracellular space through pyroptosis, a programmed cell death leading to rapid release of cellular contents such as HMGB1 and SQSTM1 [10].

## 3. Role of Cytoplasmic PRRs (Caspase-11/4/5/1) in the Regulation of Pyroptosis and HMGB1 Release

We and others demonstrated that ultra-pure LPS (free from any contaminating bacterial proteins, lipids, or nucleic acids) completely failed to induce HMGB1 release, unless the initial LPS priming was accompanied by a second stimulus (e.g., ATP) [30,31]. However, crude LPS that might carry trace amounts of bacterial proteins, lipids and nucleic acids, triggered a marked HMGB1 release [9]. It is possible that some contaminating bacterial proteins and lipids might enhance endocytosis of LPS, and consequently facilitate its innate recognition by cytoplasmic PRRs such as Casp-11/4/5. Indeed, when LPS was delivered to cytoplasmic Casp-11/4/5 either via CD14/TLR4 receptor-mediated endocytosis or bacteria-derived outer membrane vesicles (OMV) [32], it induced “non-canonical” inflammasome activation via oligomerization and proximity-induced activation of Casp-11/4/5 (Figure 1) [33]. The activated Casp-11/4/5 then catalyzes the cleavage of Gasdermin D (GSDMD) to form cytoplasmic membrane pores that cause immediate ionic gradient loss, osmotic burst and cell membrane rupture, a process aforementioned as “pyroptosis”. For the optimal activation of non-canonical inflammasome, both type I IFN-α/β and type II IFN-γ are needed to up-regulate Casp-11/4/5 [34,35] as well as guanylate-binding proteins [36] responsible for disrupting pathogen-containing vacuoles and releasing LPS. Coincidently, we and others demonstrated that LPS-inducible type I IFN-α/β [37,38] and type II IFN-γ [28] effectively stimulated innate immune cells to release HMGB1 in a time- and dose-dependent fashion.

In contrast, the “canonical” inflammasome activation is characterized by the oligomerization of intracellular “nucleotide-binding oligomerization domain (NOD)-like receptors” (NLRs such as NLRP1, NLRP3, and NLRC4) and the “apoptosis-associated speck-like protein containing a C-terminal caspase recruitment domain” (ASC) adaptor, as well as the recruitment and activation of pro-Casp-1 (Figure 1) [30]. Specifically, the pro-Casp-1 forms a heteromeric protein complex with an ASC adaptor and a NLR receptor, and the resultant protein complex, termed the “inflammasome”, is responsible for cleaving pro-Casp-1 to generate Casp-1, which triggers canonical inflammasome activation and pyroptosis via GSDMD cleavage [30]. Likewise, the optimal activation of canonical inflammasome also depends on a two-step process: (1) a priming signal elicited by extracellular PAMPs (e.g., LPS) to up-regulate NLRP3 expression; and (2) a secondary signal elicited by extracellular damage-associated molecular pattern (DAMPs, e.g., ATP) to induce NLRP3 oligomerization with ASC and pro-Casp-1 (Figure 1). Notably, the cleavage of pannexin-1 (Panx1) hemichannel by Casp-11/4/5 might be needed for releasing ATP and activating the purinergic P2X_7_ receptor (P2X_7_R) and inflammasome signalings (Figure 1) [39,40]. Consistently, we found that crude LPS also markedly up-regulated Panx1 expression in macrophages and monocytes, and consequently elevated their hemichannel activities to release ATP [41], supporting a pathogenic role of Panx1 in LPS-induced HMGB1 release and animal lethality [39] (Figure 1). 

It is thus possible that following cytoplasmic translocation, HMGB1 could be secreted extracellularly through Casp-1- or Casp-11/4/5-mediated inflammasome activation and pyroptosis (Figure 1). Recent evidence suggested that inflammasome-dependent HMGB1 release could not occur immediately after the formation of GSDMD membrane pores, but became prominent following the rupture of cytoplasmic membranes [42,43]. Consistently, pharmacological inhibition (with a broad-spectrum Caspase inhibitor Z-VAD-FMK) or genetic disruption of key inflammasome components (e.g., Casp-1 or Nlrp3) uniformly blocked the LPS/ATP-induced HMGB1 secretion [30,44]. Likewise, genetic disruption of interferon-induced double-stranded RNA-activated protein kinase (PKR) expression or pharmacological inhibition of its phosphorylation similarly reduced the LPS-induced inflammasome activation [31,45], pyroptosis [31,45], and HMGB1 release [31]. Thus, crude LPS may prime macrophages by simultaneously up-regulating PKR expression and eliciting Panx-1-mediated ATP release, thereby activating P2X_7_R [46] to induce a feed-forwarding PKR/inflammasome activation, pyroptosis and HMGB1 secretion (Figure 1). 

In addition, HMGB1 can also be passively released by somatic cells undergoing cytoplasmic membrane destruction due to accidental mechanical events or regulated processes governed by other caspases or kinases. For instance, circulating levels of HMGB1 were rapidly elevated in critical ill patients with non-penetrating trauma [47,48,49], thereby contributing to trauma-induced dysregulated inflammation, immune paralysis or immunosuppression. Even following viral infections with influenza [50,51] or SARS-CoV-2 [52], proinflammatory cytokines such as TNF and IFN-γ can also induce necroptosis [53,54,55] or PANoptosis [52] via other caspases and kinases such as the Receptor-Interacting Serine/Threonine Kinase 3 (RIPK3) [50,51] and Casp-8 [55]. Thus, various cell death pathways can potentially lead to the passive release of HMGB1 following traumatic injuries or microbial infections. However, the possible roles of HMGB1 and various other cytokines in the pathogenesis of lethal infections such as COVID-19 remain controversial, because there is still a lack of clear association between many cytokine biomarkers and the severity of viral infections [56,57].

## 4. Pathogenic Role of Extracellular HMGB1 in Dysregulated Inflammation, Immunosuppression, and Immune Paralysis

Once released, extracellular HMGB1 can bind various PRRs and PAMPs to orchestrate divergent inflammatory responses. For instance, HMGB1 can bind TLR4 [58,59,60], TLR9 [61], receptor for advanced glycation end products (RAGE) [62], cluster of differentiation 24 (CD24)/Siglec-10 [63], Mac-1 [64], or single-transmembrane-domain proteins (e.g., syndecans) [65]. Due to its relatively higher affinity to TLR4 (K_D_ = 22.0 nM) [66] and lower affinity to RAGE (K_D_ = 97.7-710 nM) [67,68], HMGB1 might first bind TLR4 when it was actively secreted by innate immune cells at relatively lower amounts [69]. Consequently, it could directly activate macrophages [70], neutrophils [71] and endothelial cells [72] to produce various cytokines and chemokines [58,72,73,74,75,76] partly through MyD88-IRAK4-dependent signaling pathways (Figure 2A).

When HMGB1 was passively released by innate immune and somatic cells at relatively higher levels, it might also bind various microbial PAMPs (e.g., CpG-DNA or LPS) and RAGE [67,77] and consequently promoted RAGE-receptor-mediated endocytosis of these microbial products (Figure 2B) [78]. Upon reaching acidic endosomal and lysosomal compartments near HMGB1′s isoelectric pH, HMGB1 became neutrally charged and set free its cargos (LPS or CpG-DNA) [78], thereby facilitating their recognition by respective PRRs such as TLR9 [61] or Casp-11 [78] to augment inflammatory responses (Figure 2B). Furthermore, the engagement of RAGE with HMGB1 might also induce chemotaxis [79] and the migration of monocytes, dendritic cells [80,81] and neutrophils [64], thereby facilitating the recruitment of innate immune cells to site of the infection to orchestrate inflammatory responses [79] (Figure 2B). Finally, the engagement of HMGB1 with RAGE [67,77] might also induce TLR4 internalization and desensitization to subsequent stimulus (e.g., endotoxin), and might even trigger macrophage pyroptosis [78,82] via a cascade of events including cathepsin B release from ruptured lysosomes followed by pyroptosome formation and Casp-1 activation (Figure 2B).

In neutrophils, HMGB1 can bind TLR4 to promote the formation of neutrophil extracellular traps (NETs), thereby amplifying neutrophil-mediated inflammatory responses [83]. In contrast, the engagement of RAGE by HMGB1 can adversely impair neutrophil NADPH-dependent production of reactive oxidation species (ROS) and associated bacterial killing, contributing to sepsis-induced immune paralysis and immuno-suppression [84,85]. Consistently, the blockade of extracellular HMGB1 activities with neutralizing antibodies even during a late stage of sepsis still restored neutrophil NADPH activity and anti-bacterial capacities [85]. Thus, excessive HMGB1 release contributes to the pathogenesis of lethal infections by posing divergent adverse effects such as immune tolerance [86,87], immune paralysis [84,85,88] and immunosuppression [85,89] (Figure 2B).

## 5. Positive Regulators of LPS-Induced HMGB1 Release

In addition to LPS-inducible type I IFN-α/β [37,38] and type II IFN-γ [28], human serum amyloid A (SAA) also effectively induced HMGB1 release by innate immune cells in a TLR4/RAGE-dependent fashion [90] (Figure 3). Consistent with its capacity in stimulating NLRP3 inflammasome activation [91,92], we observed that SAA also stimulated PKR expression and phosphorylation [90]. Conversely, pharmacological inhibition of PKR inhibited SAA-induced HMGB1 release [90], supporting an important role for PKR phosphorylation, inflammasome activation and pyroptosis in the SAA-induced HMGB1 release (Figure 1). In addition, some LPS-inducible enzymes [such as the 14 kDa type II secretory phospholipase A_2_ (sPLA_2_), inducible nitric oxide synthase (iNOS), and pyruvate kinase M2 (PKM2)] were also implicated in the regulation of LPS-induced HMGB1 release (Figure 3) [29,93,94,95,96]. In agreement with these findings, we found that human SAA effectively up-regulated the expression of sPLA_2_-IIE and sPLA_2_-V in murine macrophages (Figure 1 and Figure 3) [97], and concurrently induced HMGB1 release [90]. Conversely, the suppression of sPLA_2_-IIE expression by high density lipoproteins (HDL) also attenuated SAA-induced HMGB1 release, supporting a role of sPLA_2_ in the regulation of HMGB1 release [97]. It is not yet known whether sPLA_2_s facilitate HMGB1 release partly by catalyzing the production of lyso-phosphatidylcholine (LPC) and leukotrienes that are capable of activating NLRP3 inflammasome and pyroptosis (Figure 1) [98,99,100]. 

Finally, both crude LPS and human SAA effectively up-regulated the expression of hemichannel molecules such as Panx1 [41] and Connexin 43 (Cx43) [101] in innate immune cells (Figure 1 and Figure 3). The possible role of Cx43 in the regulation of LPS-induced HMGB1 release was supported by our findings that several Cx43 mimetic peptides, the GAP26 and Peptide 5 (ENVCYD), simultaneously attenuated LPS-induced hemichannel activation and HMGB1 release [101]. It was further supported by observation that genetic disruption of macrophage-specific Cx43 expression conferred protection against lethal endotoxemia and sepsis [102]. It is possible that Cx43 hemichannel provides a temporal mode of ATP release [103,104], which then contributes to the LPS-stimulated PKR phosphorylation, inflammasome activation, pyroptosis and HMGB1 secretion (Figure 1 and Figure 3) [41,101]. Intriguingly, recent evidence has suggested that macrophages also form Cx43-containing gap junction with non-immune cells such as cardiomyocytes [105], epithelial [106,107] and endothelial cells [108]. It is possible that innate immune cells may communicate with non-immune cells through Cx43-containing gap junction channels to regulate HMGB1 release and to orchestrate inflammatory responses [109,110]. Interestingly, recent studies have revealed an important role of lipid peroxidation [111] and cAMP immune-metabolism [112] in the regulation of Casp-11-mediated “non-canonical” inflammasome activation and pyroptosis (Figure 3). However, the possible role of these immunometabolism pathways in the regulation of LPS-induced HMGB1 release remains an exciting subject of future investigations. 

## 6. Negative Regulators of the LPS-Induced HMGB1 Release and Action

During evolution, mammals have evolved multiple negative regulatory mechanisms to inhibit HMGB1 release and action. For instance, a local feedback mechanism could be instilled by injured cells via the release of a ubiquitous biogenic molecule, spermine, which inhibited the LPS- and HMGB1-induced release of multiple cytokines and chemokines (e.g., TNF, IL-6, MIP-2, and RANTES) from macrophages and monocytes [70,113,114,115]. Notably, spermine exerted its anti-inflammatory effect in conjunction with a liver-derived negative acute-phase protein, fetuin-A (Figure 3), which served as an opsonin for the cellular uptake of cationic anti-inflammatory molecules such as spermine [116]. In an animal model of lethal endotoxemia, circulating fetuin-A levels were decreased in an anti-parallel fashion when circulating HMGB1 levels were elevated [9,117]. However, supplementation of endotoxemic animals with exogenous fetuin-A resulted in a significant reduction in circulating HMGB1 levels [117]. It is plausible that fetuin-A negatively regulated LPS-induced HMGB1 release partly by facilitating the cellular uptake of cationic anti-inflammatory molecules (spermine), and partly by stimulating macrophages-mediated ingestion and elimination of apoptotic neutrophils [118,119]. This is relevant, because inefficient elimination of apoptotic cells might adversely lead to excessive accumulation of late apoptotic and/or secondary necrotic cells, which may cause passive leakage of HMGB1 and other DAMPs [120]. 

In addition, recent evidence suggested that the central nervous system could also attenuate peripheral innate immune response through efferent vagus nerve (Figure 3) [121], which could release neurotransmitter such as acetylcholine to inactivate macrophages via nicotinic cholinergic receptors [122]. Indeed, stimulation of the vagus nerve by physical methods (e.g., electrical or mechanical) [123,124] or chemical agonists (such as nicotine and GTS-21) [125,126] conferred protection against lethal endotoxemia partly by attenuating systemic HMGB1 accumulation. Furthermore, mammals have also evolved other neuro-immune pathways by which the PTEN-induced putative kinase 1 (PINK1) and parkin RBR E3 ubiquitin protein ligase (PARK2) counter-regulate lethal systemic inflammation through another neurotransmitter, dopamine (Figure 3) [127], which could turn off systemic inflammation through suppressing NLRP3 inflammasome activation [128]. 

Finally, emerging evidence has supported a possible role of several endogenous proteins such as thrombomodulin [129], haptoglobin [130], complement factor 1q (C1q) [131], heat shock protein 70 (HSP70) [132,133], vasoactive intestinal peptide [134], urocortin [135], and ghrelin [136] in the regulation of LPS-induced HMGB1 release or cytokine activities. For instance, an endothelial anticoagulant cofactor, thrombomodulin, could bind HMGB1 to prevent its interaction with macrophage cell-surface receptors [137], thereby preventing HMGB1-induced inflammatory responses [129,138]. Similarly, a liver-derived acute-phase protein, haptoglobin (Hp, Figure 3), could capture HMGB1 to trigger CD163-dependent endocytosis of HMGB1/Hp complexes, and induced the production of anti-inflammatory enzymes (heme oxygenase-1) and cytokines (e.g., IL-10) [69,130]. In addition, a component factor 1q (C1q) capable of binding antigen-antibody complexes to initiate the classical complement pathway [15], also interacted with HMGB1 (K_D_ = 200 nM) and formed a tetramolecular complex with RAGE and LAIR-1, resulting in the production of anti-inflammatory cytokines (e.g., IL-10) and pro-resolution lipid mediators [131,139]. Similarly, an anticoagulant polysaccharide, heparin, or other chemically modified (2-O, 3-O desulfated) heparins, could all bind HMGB1 and prevented its interaction with LPS [140] or RAGE receptor [67], thereby inhibiting Casp-11-mediated inflammasome activation and pyroptosis, as well as HMGB1-mediated immunosuppression [141]. Thus, in sharp contrast to exogenous PAMPs (e.g., CpG-DNA and LPS), many endogenous proteins and polysaccharides could bind HMGB1 to tilt the balance towards anti-inflammatory responses via distinct signaling pathways [130,131,137,139,140].

## 7. Pharmacological Modulation of LPS-Induced HMGB1 Release or Action

Our seminal discovery of HMGB1 as a late mediator of lethal endotoxemia has stimulated extensive interest in search for HMGB1-targeting pharmacological inhibitors ranging from small molecules to large biological agents.

### 7.1. Small-Molecule Inhibitors

Among many medicinal herbs that we screened for possible HMGB1-inhibiting activities, we found that aqueous extracts of Danggui (*Angelica sinensis*) [142], Gancao (*Radix glycyrrhizae*) [143], Green tea (*Camellia sinensis*) [144], and Danshen (*Saliva miltorrhiza*) [145] conferred significant protection against lethal endotoxemia partly by inhibiting LPS-induced HMGB1 release via dramatically distinct mechanisms. For instance, a major component of Gancao, glycyrrhizin (GZA), could directly bind HMGB1 [146] to disrupt its engagement with RAGE receptor [147], thereby conferring protection against lethal endotoxemia by inhibiting HMGB1-mediated inflammation (Figure 4) [148]. A chemical derivative of the GZA, carbenoxolone (CBX), however, dose-dependently inhibited the LPS-induced HMGB1 secretion [143] partly by inhibiting the LPS-induced PKR expression and phosphorylation (Figure 4). Given its capacity in inhibiting macrophage Cx43 and Panx1 hemichannel activities [149,150], CBX could also counter-regulate HMGB1 release through inhibiting LPS-induced activation of Cx43 or Panx1 hemichannels in innate immune cells (Figure 4) [143].

A major green tea component, EGCG, prevented the LPS-induced HMGB1 release strategically by destroying it in the cytoplasm via a cellular degradation process—autophagy (Figure 4) [151]. Specifically, EGCG could be trafficked into cytosol to conjugate with cytoplasmic HMGB1 either covalently with the free thiol group of cysteine residues [152] or non-covalently via hydrogen bonding, aromatic stacking or hydrophobic interactions [153]. Consequently, EGCG induced the formation of EGCG–HMGB1 complexes that were eventually engulfed by double-membraned autophagosomes, and subsequently degraded by acidic lysosomal hydrolases [151].

In contrast, a derivative of major ingredient of Danshen, tanshinone IIA sodium sulfonate (TSN-SS, Figure 4), selectively inhibited LPS-induced HMGB1 release without affecting the secretion of other cytokines and chemokines (such as IL-6, IL-12p40/p70, KC, MCP-1, MIP-1α, MIP-2, and TNF) [145]. Unlike EGCG, TSN-SS itself was unable to stimulate autophagic HMGB1 degradation, but instead facilitated the endocytosis of extracellular HMGB1 through clathrin- and caveolin-dependent endocytosis (Figure 4) [154]. Because cytoplasmic HMGB1 could induce autophagy [155,156,157], the TSN-SS-mediated HMGB1 endocytosis may be paralleled with the occurrence of HMGB1-induced autophagy, and eventually converged on a lysosome-dependent final common pathway that eventually leads to HMGB1 degradation (Figure 4). That is, the HMGB1-containing endosomes might fuse with HMGB1-induced autophagosomes to form amphisomes [158,159], and then merge with lysosomes to trigger HMGB1 degradation via a lysosome-dependent pathway (Figure 4) [154]. Given its demonstrated safety in China as a medicine for patients with cardiovascular disorders, and its capacity to inhibit HMGB1 release after LPS stimulation, TSN-SS may be a promising therapeutic agent for inhibiting HMGB1 release in clinical settings [27].

### 7.2. Development of Antibodies Targeting HMGB1 or Its in-Crime Binding Partners

Neutralizing antibodies against endotoxin [160] or early cytokines (e.g., TNF) [161,162] were protective in an animal model of lethal endotoxemia, but unfortunately failed in clinical trials [163,164,165]. This failure partly reflects the complexity of the underlying lethal infections, and the associated heterogeneity of the patient populations [166]. In contrast to the relatively unified age, body weight, and genetic background of experimental animals in pre-clinical studies, patients recruited in clinical studies usually exhibit intrinsic (genetic and epigenetic) heterogeneity, and harbor various underlying comorbidities that complicate the pathogenesis and progression of clinical sepsis. Nevertheless, the investigation of pathogenic cytokines (such as TNF) has led to the development of anti-TNF therapy for patients with debilitating chronic inflammatory diseases, such as rheumatoid arthritis [167]. Accordingly, we have generated polyclonal and monoclonal antibodies against human HMGB1 and tested their efficacy in animal models of lethal endotoxemia and sepsis induced by a surgical procedure termed cecal ligation and puncture (CLP). In an animal model of lethal endotoxemia, HMGB1-specific polyclonal antibodies were protective in a dose-dependent fashion [9]. In an animal model of CLP-induced sepsis, HMGB1-neutralizing monoclonal antibodies (mAbs) [44,168] conferred significant protection even when the first dose was given 24 h after disease onset [23,169,170,171], establishing HMGB1 as a “late” mediator of experimental sepsis with a relatively wider therapeutic window than that offered by early proinflammatory cytokines.

As aforementioned, many endogenous proteins (such as thrombomodulin, haptoglobin, and C1q) [129,130,131] could physically interact with HMGB1, promoting the search for other HMGB1-binding proteins that might also affect its biological functions. During this process, we noticed that the blood level of a 20 kDa protein was almost completely depleted in patients who died of sepsis. This 20 kDa protein was identified as human tetranectin (TN) by mass spectrometry and immunoblotting assays [172]. Intriguingly, TN selectively inhibited the LPS- and SAA-induced HMGB1 release without affecting the parallel release of other cytokines and chemokines [172], partly because TN could capture extracellular HMGB1 and facilitated the endocytosis of TN/HMGB1 complexes, thereby enhancing HMGB1-induced pyroptosis (Figure 5) [172].

As discussed earlier, pyroptosis not only allows excessive release of HMGB1 and SQSTM1 that adversely drive a life-threatening dysregulated inflammatory response to lethal infections, but also leads to immune cell depletion and possible immunosuppression that may have compromised the host innate immunity against lethal infections (Figure 5). Accordingly, we have developed a panel of TN-specific mAbs that effectively prevented both harmful HMGB1/TN interaction and resultant macrophage pyroptosis and lethal sepsis [172]. It suggested that TN domain-specific mAbs may confer protection against lethal sepsis partly by preventing harmful TN/HMGB1 interaction that may adversely trigger macrophage pyroptosis and immunosuppression (Figure 5) [173,174]. This antibody strategy has also suggested a possibility to develop therapeutic antibodies against harmless proteins colluding with sepsis mediators [173,174,175].

Surprisingly, we recently discovered that two TN-reactive mAbs capable of rescuing mice from lethal sepsis also cross-reacted with the human ACE2 receptor binding motif (RBM) of SARS-CoV-2 (Figure 5), with an estimated K_D_ of 17.4 and 62.8 nM, respectively [176]. The estimated K_D_ was comparable to that of other SARS-CoV-2 RBD-binding neutralizing antibodies (K_D_ = 14–17 nM) derived from COVID-19 patients [177]. Furthermore, these TN/RBM-reactive mAbs competitively inhibited RBM-ACE2 interactions in vitro [176], and selectively impaired the RBM-induced secretion of the granulocyte macrophage colony-stimulating factor (GM-CSF) [176]. Our findings fully supported the emerging notion that GM-CSF might be a key biomarker for SARS-CoV-2-induced cytokine storm in a subset of COVID-19 patients with more severe pneumonia often escalating to respiratory failure and death [178,179,180,181,182] (Figure 5), although the possible roles of GM-CSF and other cytokines in the pathogenesis COVID-19 remain a subject of ongoing debate [56,57]. Nevertheless, it is possible that these TN/RBM-reactive mAbs might selectively prevent its interaction with a receptor involved in the GM-CSF induction, but did not interfere with its engagement with other pattern recognition receptors responsible for the induction of other cytokines or chemokines [183]. Because HMGB1 was similarly accumulated in patients with COVID-19 [184], these TN/RBM-reactive mAbs might simultaneously block harmful TN/HMGB1 interaction and resultant immunosuppression, and suppress possible SARS-CoV-2/ACE2 interaction to inhibit GM-CSF production in patients with COVID-19 and other lethal infections [71].

## 8. Future Perspectives

Microbial infections and sepsis remain a major clinical problem that accounts for approximately 20% of total deaths worldwide [1], and annually costs more than $62 billion in the U.S. alone [185]. Despite a robust increase in the understanding of the pathophysiology of sepsis, many antibody-based strategies targeting early cytokines (such as TNF or IL-1) failed in clinical settings. Currently, there is still no effective therapy [173] other than adjunctive use of antibiotics, fluid resuscitation, and supportive care [185]. Thus, it would be beneficial to test the therapeutic efficacy of some promising HMGB1 inhibitors in clinical settings. For instance, a selective HMGB1 inhibitor, TSN-SS, has already been used in China as a medicine for patients with cardiovascular disorders. The dual effects of TSN-SS in attenuating late inflammatory response and improving cardiovascular functions make it a promising therapeutic candidate for treating lethal infections. Similarly, it would also be exciting to test the therapeutic efficacy of TN-specific mAbs that effectively prevented its undesired interaction with pathogenic mediators (HMGB1) and resultant immunosuppression [172]. The discovery of mAbs capable of disrupting TN/HMGB1 interaction and endocytosis and rescuing animals from lethal sepsis has suggested an exciting possibility to develop therapeutic antibodies against harmless proteins colluding with disease mediators [175]. Given the cross-reactivity of several TN-reactive monoclonal antibodies to the RBM of SARS-CoV-2, it will be extremely important to test the efficacy of these TN/RBM-reactive monoclonal antibodies in clinical trials of COVID-19 and other microbial infections.

## Figures and Tables

**Figure 1 cells-10-02220-f001:**
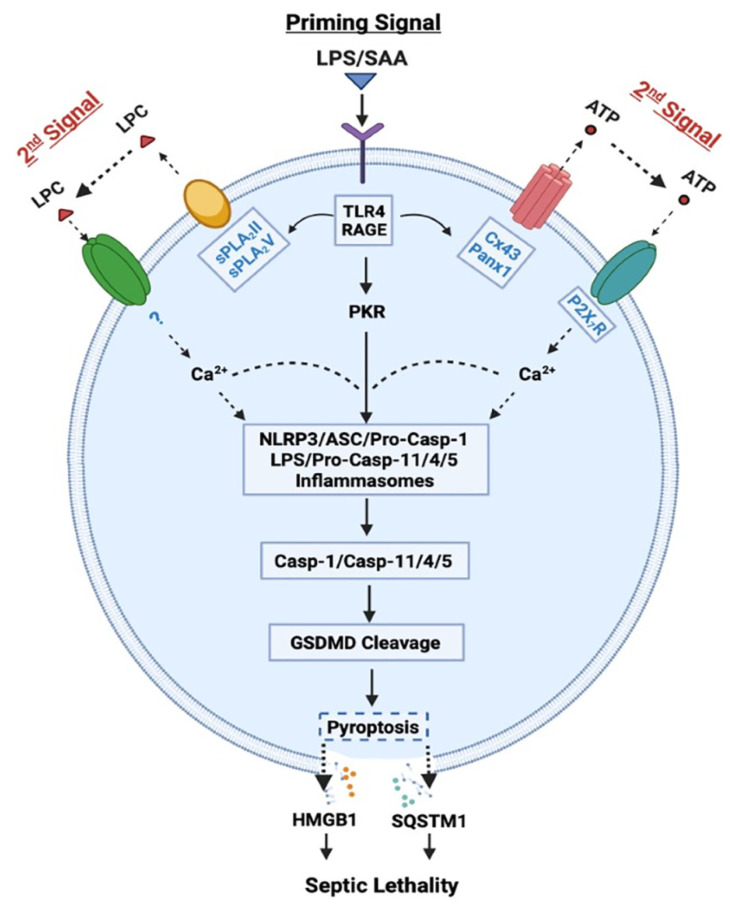
Role of Casp-1-mediated canonical and Casp-11/4/5-mediated non-canonical inflammasome activation in LPS- or SAA-induced pyroptosis and HMGB1 release. LPS or SAA may prime innate immune cells to up-regulate the expression of Cx43/Panx1 hemichannels, sPLA_2_s and interferon-induced double-stranded RNA-activated protein kinase (PKR), thereby eliciting the release of ATP or LPC that may activate P2X_7_R- or other receptor-mediated Ca^2+^ signaling. It then induces a feed-forwarding activation of PKR and inflammasome, cleavage of GSDMD, pyroptosis, and subsequent release of late mediators (such as HMGB1 and SQSTM1) of lethal infections.

**Figure 2 cells-10-02220-f002:**
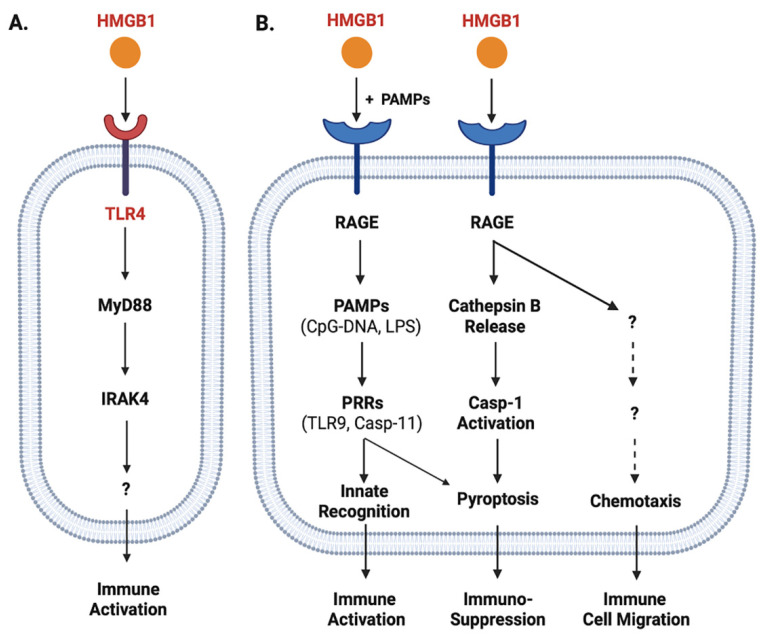
Role of TLR4 and RAGE in the regulation of HMGB1-mediated divergent inflammatory responses. HMGB1 can bind different PRRs such as TLR4 (Panel **A**) and RAGE (Panel **B**) with different affinities, and consequently induce divergent inflammatory responses such as immune cell migration, immune activation, or pyroptosis and resultant immunosuppression.

**Figure 3 cells-10-02220-f003:**
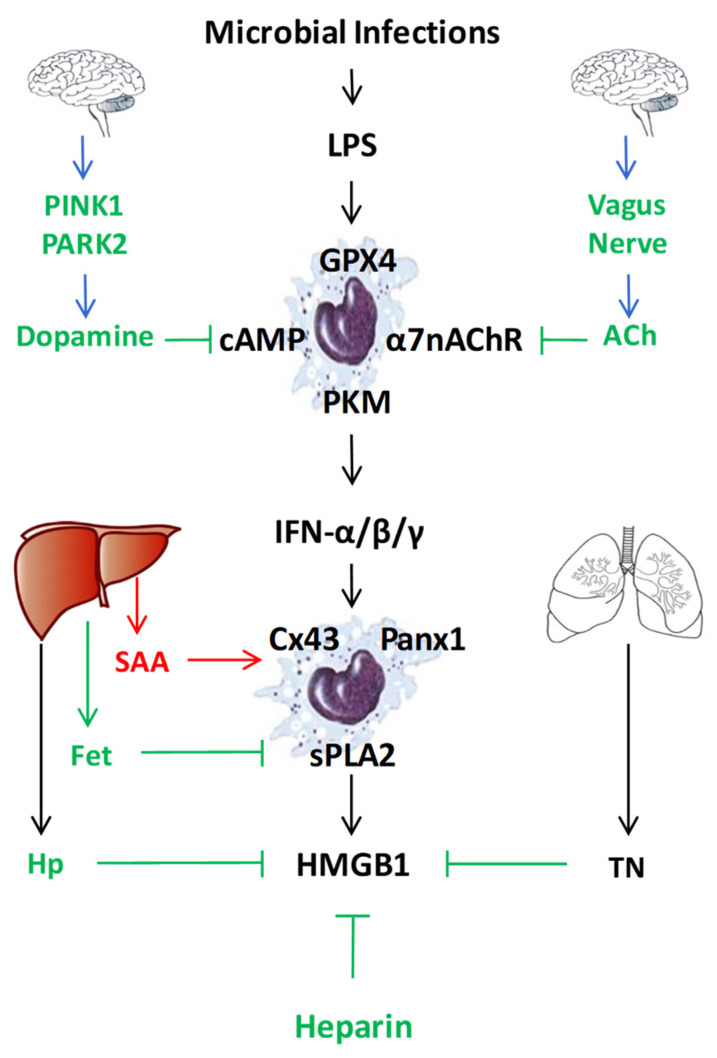
Endogenous regulators of LPS-induced HMGB1 release or action. To regulate the LPS-induced HMGB1 release or action, mammals have evolved multiple regulatory mechanisms that include neuro-immune pathways, liver-derived acute-phase proteins (e.g., SAA, Fetuin-A (Fet), Haptoglobin (Hp)), as well as other endogenous proteins (e.g., tetranectin (TN)) or polysaccharides (heparin).

**Figure 4 cells-10-02220-f004:**
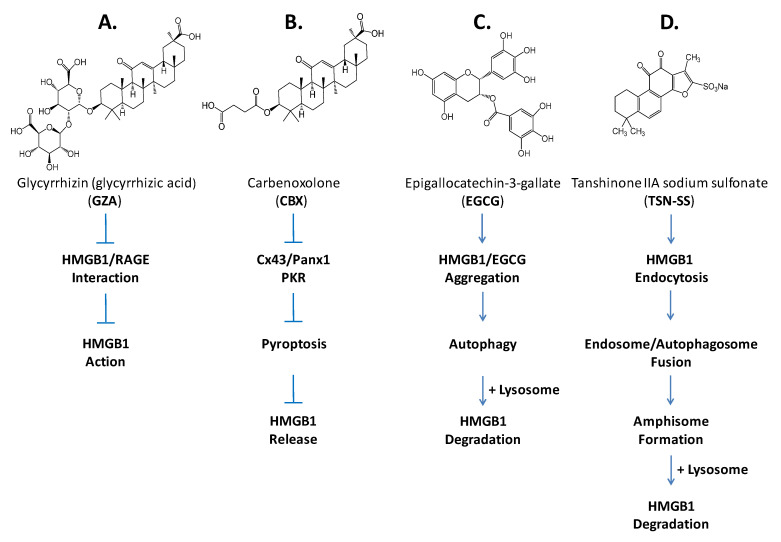
Distinct mechanisms of several pharmacological inhibitors of HMGB1 release or action. Different herbal compounds or derivatives inhibit HMGB1 release or action through distinct mechanisms that include: (**A**) direct binding to inhibit its engagement with various PRRs; (**B**) direct binding to induce its aggregation and autophagic degradation; (**C**) inhibition of key signaling molecules (PKR and hemichannels) involved in inflammasome activation and pyroptosis; and (**D**) induction of its endocytosis and lysosome-dependent degradation.

**Figure 5 cells-10-02220-f005:**
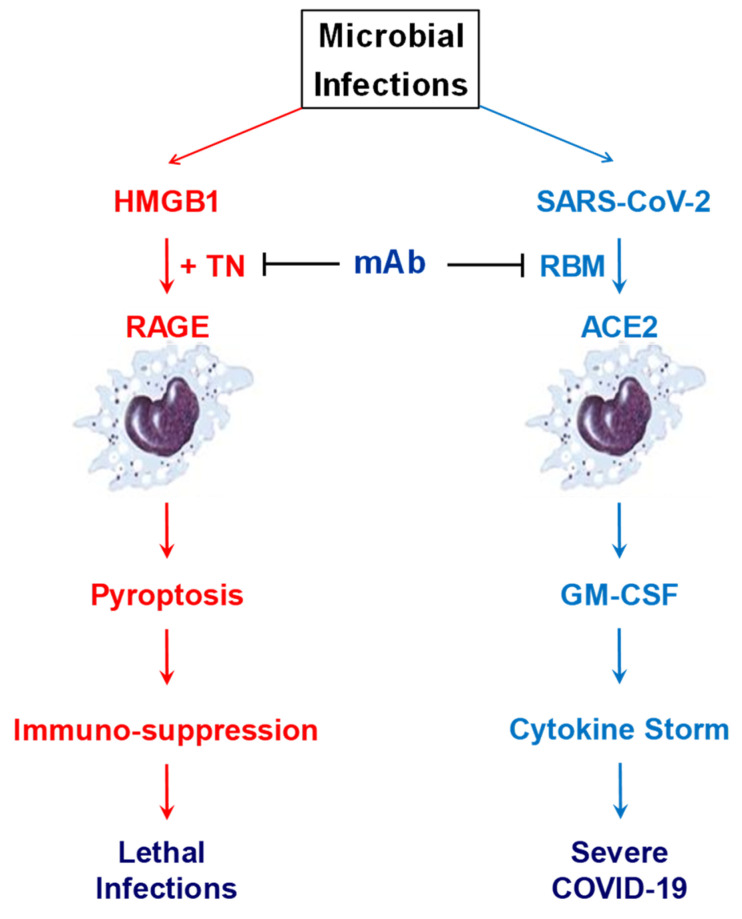
Potential therapeutic effects of cross-reactive monoclonal antibodies against TN and RBM of SARS-CoV-2. Some TN domain (NDALYEYLRQ)-specific mAbs may confer protection against lethal infections partly by preventing harmful TN/HMGB1 interaction that may adversely trigger macrophage pyroptosis and immunosuppression. Some TN-reactive mAbs also cross-reacted with a tyrosine (Y)-rich segment (YNYLYR) in the RBM of SARS-CoV-2, and specifically inhibited RBM-induced production of GM-CSF, a biomarker and mediator of COVID-19. The dual effects of these cross-reactive mAbs in attenuating immuno-suppression and SARS-CoV-2-induced GM-CSF production make them promising therapeutic candidates for treating COVID-19 and other lethal infections.

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
