# Peer review of "Endogenous Regulation and Pharmacological Modulation of Sepsis-Induced HMGB1 Release and Action: An Updated Review"

_cells, 2021, doi:10.3390/cells10092220_

Round 1

Reviewer 1 Report

In this review article authors articulate the mechanistical aspects of HMGB1 in various cellular processes and its potential applications  in resolving dysregulated immune responses during host pathogen interactions. Authors are expert in the field and they have appropriately cited their previous research work in this manuscript. The manuscript is well written however, following are few comments which should be considered. 

  1. The High-mobility group protein box-1 also play important role as alarmins which is relevant trauma induced sepsis. Author have cited some of their work related to cecal ligation and puncture so it may be relevant to cite some of the work to highlights its physiological relevance and significance.
  2. HMGB1 also play important role in cell recruitment and diapedesis. Neutrophil paralysis play important role in the progression of sepsis so author should consider discussing this aspect.  
  3. Authors discussed some of the aspects of HMGB1 in the pathophysiology of COVID19 which is interesting. However, the cytokine storm is a controversial topic in COVID19 because some of the hallmark cytokines are not playing important role during the disease progression. Author should cite some of the relevant original research article for a balance opinion on this subject. 

Author Response

Response to Reviewer #1

In this review article authors articulate the mechanistical aspects of HMGB1 in various cellular processes and its potential applications  in resolving dysregulated immune responses during host pathogen interactions. Authors are expert in the field and they have appropriately cited their previous research work in this manuscript. The manuscript is well written however, following are few comments which should be considered. 

We thank the reviewer for these positive expert comments, and have revised the manuscript as per the reviewer’s specific comments below.

  1. The High-mobility group protein box-1 also play important role as alarmins which is relevant trauma induced sepsis. Author have cited some of their work related to cecal ligation and puncture so it may be relevant to cite some of the work to highlights its physiological relevance and significance.

We thank the reviewer for the comment, and have cited three new references regarding systemic HMGB1 release in critical ill trauma patients (Reference 47-49, Page 4, Line 200-205).

  1. HMGB1 also play important role in cell recruitment and diapedesis. Neutrophil paralysis play important role in the progression of sepsis so author should consider discussing this aspect.  

We have added a paragraph to discuss the important impact of HMGB1 on neutrophil dysfunctions in the revised manuscript (Reference 83-85; Page 6, Line 335-345).

  1. Authors discussed some of the aspects of HMGB1 in the pathophysiology of COVID19 which is interesting. However, the cytokine storm is a controversial topic in COVID19 because some of the hallmark cytokines are not playing important role during the disease progression. Author should cite some of the relevant original research article for a balance opinion on this subject. 

We have added a brief discussion on the possible mechanisms of HMGB1 release in response to viral infections (e.g., influenza and COVID-19, Reference 50-55, Line 205-210), as well as the possible role of HMGB1 and other cytokines in the pathogenesis of COVID-19 (Reference 56-57; Page 4, Line 210-213; and Page 11, Line 759-761).

Reviewer 2 Report

The review is interesting and well-written. I have some suggestions to improve its quality.

I suggest to the Authors to mention the word sepsis in the title.

The manuscript must be fully revised for English.

Introduction.

  • In the first sentence,  Authors should provide a citation.
  • The second sentence start as follows: “Its pathogenesis..” The Authors refer to sepsis I suppose but they should specify it.
  • I suggest to remove the following sentence or substitute it: “ In parallel with a few recently invited reviews [14,15],”

Figures 1 and 2 are of low resolution. Please, provide figures at high resolution.

Author Response

Response to Reviewer #2

The review is interesting and well-written. I have some suggestions to improve its quality.

We thank the reviewer for the positive comments. 

I suggest to the Authors to mention the word sepsis in the title.

We have now mentioned the word sepsis in the title.

The manuscript must be fully revised for English.

We have carefully proof-read the entire manuscript for errors in syntax and styling, and made numerous edits accordingly.

Introduction. In the first sentence,  Authors should provide a citation.

We have provided a reference to the first sentence of the introduction (Reference 1; Page 1, Line 29) after making minor revision to the second half of this sentence.  .

The second sentence start as follows: “Its pathogenesis..” The Authors refer to sepsis I suppose but they should specify it.

We thank the reviewer for the comment, and have revised this sentence to make it clearer (Page 1, Line 29).

I suggest to remove the following sentence or substitute it: “ In parallel with a few recently invited reviews [14,15],”

We have revised this sentence to reflect the fact that our reviews intended to “complement two relevant reviews in this special issue” (Page 2, Line 78), which was entitled “Regulation of HMGB1 Release in Health and Disease” and edited by the senior author, Dr. Haichao Wang (as the Guest Editor).

Figures 1 and 2 are of low resolution. Please, provide figures at high resolution.

We have provided modified Figure 1 and 2 with enhanced resolution.
